# The Impact of Environmental Regulation on Agricultural Productivity: From the Perspective of Digital Transformation

**DOI:** 10.3390/ijerph191710794

**Published:** 2022-08-30

**Authors:** Zhiqiang Zhou, Wenyan Liu, Huilin Wang, Jingyu Yang

**Affiliations:** 1School of Business, Hunan University of Science and Technology, Yuhu District, Xiangtan 411201, China; 2School of Metallurgy and Environment, Central South University, Yuelu District, Changsha 410083, China; 3International College, National Institute of Development Administration, 118 Moo3, Sereethai Road, Klong-Chan, Bangkapi, Bangkok 10240, Thailand; 4Department of Medical Bioinformatics, University of Göttingen, 37077 Göttingen, Germany

**Keywords:** voluntary environmental regulation, total factor productivity, digital transformation, digital agriculture

## Abstract

China’s goal of becoming a strong agricultural country cannot be achieved without the modernization and digital transformation of the agricultural sector. Presently, China’s agriculture has ushered in the era of digital economy transformation. The digital transformation of agriculture has played a huge role in improving agricultural productivity, promoting sustainable development of China’s agricultural economy, and achieving sustainable development goals. The deep integration of digital economy and agricultural economy has become an important issue of The Times. This study uses a two-way fixed-effects model and an instrumental variable method to examine the impact of environmental regulation on agricultural total factor productivity. Using the method of mechanism analysis, the conduction path of improving agricultural productivity under the means of environmental regulation is discussed. Therefore, the visualization analysis results based on the panel data of Chinese agricultural enterprises from 2011 to 2019 show that the distribution of digital transformation and productivity level of enterprises is uneven and tends to be stable in space. The empirical analysis results show that there is a direct and significant positive relationship between voluntary environmental regulation and agricultural total factor productivity. The results of mechanism analysis show that, under the means of environmental regulation, digital transformation plays an indirect role in improving agricultural productivity. On the basis of enriching and deepening the theoretical extension of the “Porter Hypothesis”, this study subtly incorporates environmental regulation, digital transformation, and agricultural productivity into a unified framework, expanding existing research.

## 1. Introduction

Agriculture is the primary condition for all production activities and an important channel for providing grain [1], non-staple food, industrial raw materials, capital, and export materials to other sectors of the national economy. Improving agricultural production efficiency is a prerequisite for improving the lives of residents [2], and it is also the basis for ensuring national independence. Agricultural development depends on agricultural environment, and a good agricultural production mode is conducive to maintaining the stability of agricultural environment and promoting the sustainable development of agriculture [3]. However, China’s huge population pressure and rapid urbanization have caused the overuse of land resources [4]. In addition, the long-term abuse of chemical fertilizers and pesticides has caused serious damage to the agricultural environment and low agricultural productivity [5,6], hindering the sustainable development of agriculture. Therefore, improving the agricultural environment is important in improving agricultural productivity and avoiding a food crises [7].

The rapid development of industrialization and urbanization, while improving productivity, has caused great pressure to the ecological environment, especially the increasingly serious soil pollution [8]. China’s Ministry of Ecology and Environment indicates that soil pollution in China has been formed over a long period of time in the course of economic and social development, among which agricultural production is an important cause of soil pollution in cultivated land. For example, sewage irrigation, fertilizers, pesticides, agricultural film, and other agricultural inputs, as well as livestock and poultry breeding, have caused soil pollution in cultivated land [9]. According to a nine-year survey of soil pollutants by China’s Ministry of Environmental Protection and Ministry of Land and Resources, 19.4 percent of farmland is being polluted. Cadmium, mercury, arsenic, copper, lead, nickel, and other heavy metals were the main pollutants, with excess rates of 7%, 1.6%, 2.7%, 2.1%, 4.8%, and 1.9% [10,11]. This poses a great threat to both food security and public health, and seriously affects land utilization and sustainable development.

Faced with serious soil pollution, low land utilization rates, and poor sustainable land development capacity, the Chinese government is actively looking for measures to alleviate the above problems. In 2016, The State Council issued the Soil Pollution Prevention and Control Action Plan, marking the beginning of China’s soil pollution control [12]. Only in 2018 did the Standing Committee of the 13th National People’s Congress pass China’s first special law on soil pollution, the Soil Pollution Prevention and Control Law [10]. By 2021, the country’s soil environmental risks were mostly under control, and the trend of worsening soil pollution was initially curbed. The safe utilization rate of polluted farmland has been kept at over 90 percent, and the soil environment of agricultural land has been generally stable. However, the average grade of the overall cultivated land quality was still not high, and the cultivated land area of medium and low land accounted for 68.76% of the total cultivated land area [13]. Therefore, the soil environmental quality of China still has enormous space for improvement, and it is necessary to further explore the new way of preventing soil pollution.

Rapid urbanization and urban area expansion have led to a sharp reduction in the number of villages and depletion of cultivated land resources [14]. By 2019, China’s cultivated land area was only 1.918 billion mu, with a per capita cultivated land area of only 1.36 mu, less than 40% of the world’s per capita level [15]. Therefore, “saving and intensive land use, strictly guarding the cultivated land red line” is an important decision to protect cultivated land and support economic and social development. It also demonstrates that in the face of the depletion of arable land resources and serious soil pollution, improving agricultural productivity and land utilization rates is the key to improving China’s agricultural competitiveness.

China’s basic national conditions of “large population and small land” [16] determine that we should realize intensive agricultural production and increase agricultural productivity as soon as possible, so as to avoid food problems [17]. Moreover, the revolutionary transformation of the agricultural sector, which is the foundation of the national economy, is crucial to achieving China’s sustainable development goals and carbon neutrality goals [18]. With the rapid development of the digital economy and becoming the main driving force of the world’s economic development, China’s agricultural digital economy shows great potential compared with other industries [19]. Therefore, accelerating agricultural digital transformation and developing digital agriculture is not only an important way to improve agricultural production and agricultural environment, but also an important strategic means to realize the sustainable development of China’s agricultural economy.

In recent years, the rapid development of the digital economy represented by digital information technologies such as the Internet, big data, and artificial intelligence has provided new ideas for improving agricultural production efficiency [20]. Since the 18th National Congress of the Communist Party of China, policy documents such as the Outline of the Digital Rural Development Strategy, the Outline of Action for Promoting the Development of Big Data, and the Implementation Opinions on Promoting the Development of Big Data in Agriculture and Rural Areas have been promulgated successively. Provides effective direction guidance [21,22,23]. Especially in 2020, the “Digital Agriculture and Rural Development Plan (2019–2025)” jointly issued by the Ministry of Agriculture and Rural Affairs and the Office of the Central Cyber Security and Informatization Committee made a comprehensive deployment on how to develop digital agriculture and promote the modernization of agriculture and rural areas [24]. The digital transformation of agriculture is starting to pay off. For example, in 2018, China’s agricultural digital economy accounted for 7.3 percent of the industry’s added value [25]. By 2020, the potential market size of smart agriculture was expected to increase to 200 billion yuan [25]. By the second half of 2021, China had completed nine agricultural iot (the internet of things) demonstration provinces and 100 digital agriculture pilots, realizing and deepening the integration of BDS and smart agricultural production, and promoting the in-depth development of agricultural e-commerce. However, compared with other developed countries, the development of digital agriculture in China is still at an early stage [18]; the level of technological innovation in digital agriculture is insufficient, and digital technology has not been integrated with actual production activities. This has caused some agricultural digital constructions to be “just in name”, only increasing but not increasing income. Especially compared to how private farms are run in the US and digital farming in Israel [26,27], China still has a long way to go.

After combing and studying relevant literature, this study finds that few scholars pay attention to the impact of environmental regulation in agriculture on agricultural productivity and land utilization rate, and few scholars discuss the mechanism of digital transformation. Therefore, different from the focus of previous studies, this study focuses on the development of digital agriculture and explores the impact of environmental regulations on agricultural productivity from the perspective of agricultural enterprises. The objectives of this study are: (1) to understand the current situation of voluntary environmental regulation (VER) and agricultural productivity of agricultural enterprises in China; (2) to analyze the impact of voluntary environmental regulation on agricultural productivity, and further study the potential role of digital transformation; (3) to provide relevant institutions (e.g., enterprises, government agriculture departments) with targeted reference opinions. Therefore, this study uses text mining and principal component analysis to determine the digital transformation index of enterprises, and then uses a two-way fixed-effects model and instrumental variable method to study and verify the impact of environmental regulation on agricultural productivity. In addition, from both theoretical and empirical analysis, this study has proved the internal conduction of digital transformation, enriching and expanding the theoretical connotation of the “Porter Hypothesis”.

Therefore, this study attempts to examine the impact of VER on agricultural productivity based on the perspective of digital transformation. From the perspective of the transmission path, we believe that the impact of VER on agricultural productivity can be separated into two action paths.

First of all, based on the Porter hypothesis, environmental regulation will trigger the compensation effect of innovation and thus, improve total factor productivity (TFP) [28]. According to the theory of planned behavior [29], intention is the motive that leads the micro subject to perform certain actions. Generally speaking, the more obvious and intense the behavior intention produced by the subject, the greater the possibility of action execution. Therefore, this study argues that the intention of enterprises to voluntarily regulate the environment will drive them to actively seek the path of innovation and productivity improvement. Therefore, we hypothesize that VER can improve agricultural productivity (H1).

Secondly, digital transformation is an important strategic means to achieve the Sustainable Development Goals [30]. In the context of digital transformation, agricultural transformation can successfully cope with various problems and challenges to ensure sustainable agricultural development [31]. On the other hand, VER and digital transformation both show pro-environment characteristics. VER of enterprises will surely promote them to change the mode of production and management and adopt pro-environmental behaviors. Therefore, this study further hypothesizes that VER improves agricultural productivity through digital transformation (H2).

The structure of this paper is as follows: Section 2 reviews relevant literature research on VER, digital transformation and TFP, and proposes research hypotheses. Section 3 introduces the related research methods and builds the test model. Section 4 introduces the data sources and reports the results of descriptive analysis of key variables. Section 5 gives the corresponding empirical test results, including benchmark regression results, robustness test results, endogeneity test results, and mechanism test results, and analyzes the reasons for the research results. Section 6 discusses the research results in detail, and puts forward corresponding policy recommendations, and also summarizes the shortcomings of this research.

## 2. Theoretical Framework

### 2.1. VER and TFP

The academic community has not reached a consistent conclusion regarding the research on the relationship between environmental regulation and TFP. In summary, there are three points of view. The first view is the “Conditioning Hypothesis” [32,33]. This view holds that the environmental regulation of enterprises to improve environmental quality will increase their “compliance costs”, resulting in a lack of funds for other investment activities, such as business operations and innovation, hindering technological progress, reducing the competitiveness of enterprises, thus offsetting the positive effect, which is brought about by environmental protection, eventually leading to a decline in TFP. The second view is the “Porter Hypothesis”. Appropriate and reasonable environmental regulation can promote enterprises to carry out technological innovation activities [34], improving the productivity of enterprises [35], and through the “innovation compensation effect” [28] enhance the competitiveness of enterprises, thereby partially or completely offsetting the cost of environmental protection, forming a “win-win” pattern of environmental protection and TFP. The third view is the “uncertain hypothesis”. This view holds that the relationship between environmental regulation and TFP is not a simple linear relationship, and the relationship between the two is complex [36]. There are many potential factors affecting the TFP of enterprises, and it is impossible to determine that environmental regulation is a key influencing factor, and the strength of different environmental regulations and the effects of transmission mechanisms are also uncertain.

In general, most studies on the relationship between environmental regulation and firms over the past 20 years are still based on the perspective of mandatory environmental regulation. For example, the studies [34,37] show that “weak” environmental regulatory measures have a better effect on the vitality of corporate innovation activities. At the same time, many foreign scholars have noticed the existence of VER by enterprises [38] and researched it. However, compared with foreign countries, Chinese scholars have completed less research on VER and its possible impact and started late, especially since the relationship between VER and TFP research is rare. From the perspective of corporate innovation and environmental performance, only a few scholars have found that VER can significantly promote corporate innovation performance [39] and ecological efficiency [40]. In fact, as the two main bodies of environmental regulation, there is a balance between the government and enterprises. As far as the government is concerned, the government closely links environmental management system certification or environmental protection qualification certification with the environmental performance of enterprises [41,42], which is to expect enterprises to achieve environmental management system certification and have certain requirements, then encourage them to be willing and active to take responsibility for environmental protection and reduce environmental pollution [43]. For enterprises, to obtain government policy support and corresponding resources, financial support [44] and to establish a good corporate image and reputation they will actively respond to the call of the government’s environmental regulation policy, send a signal to all sectors of society that they have a strong willingness and capability for sustainable development [45], and take the initiative to take some VER behaviors. For example, to accelerate the realization of the green production process and strengthen terminal governance capabilities [46], to obtain the recognition of the government, consumers and investors, and then through the obtained “order commitment”, it can make up for the expenditure of the cost of technological innovation to improve environmental protection capabilities. It is worth noting that this technological innovation process will inevitably improve the TFP of enterprises by improving resource utilization [47]. Therefore, this study proposes the following hypothesis:

**Hypothesis 1** **(H1):**
*VER has a positive effect on the TFP of enterprises.*


### 2.2. The Impact of Digital Transformation on the Relationship between Environmental Regulation and TFP

In recent years, China’s digital agricultural technology has developed rapidly [48], especially the rise and wide application of Internet of Things technology. Several practical digital agricultural technology products have been developed, and a dedicated network digital agricultural technology platform has been established and improved. Production is more efficient. The digital transformation of agricultural production is an inevitable trend of agricultural modernization moving towards a higher stage. It represents that innovation has effectively promoted the development of agricultural and rural modernization, and it is also a sign that China is moving from a largely agricultural country to a powerful agricultural country.

Digital transformation shows obvious pro-environmental characteristics [49], which not only fits the current green low-carbon life, green production, and sustainable development [50,51], it also gives an effective way of how to use limited resources correctly and efficiently to avoid environmental problems [52]. It is changing people’s traditional consumption behavior and concept. More consumers are joining the ranks of green consumption and green life. This drives enterprises to use accurate data analysis technology to capture customer needs, innovate products and services so as to meet consumer preferences, and achieve personalized customized production [53] so as to improve corporate competitiveness and enhance corporate efficiency. At the same time, business managers and policymakers have changed the concept of focusing on economic growth and tend to seek effective energy conservation and emission reduction strategies [54], improve public influence, and establish a good brand image. In terms of production, the widespread application of digital information technologies, such as artificial intelligence, blockchain, and cloud computing in enterprises has prompted a large number of enterprises, especially those with high energy consumption and high pollution [55], to transform their resources to consumption-based production and development, create green production processes, improve productivity and production efficiency, achieve energy conservation, emission reduction and green upgrades, and reduce environmental pollution.

The digital transformation of agriculture is the only way to promote the green and sustainable development of agriculture [56]. It can minimize input, reduce costs, and ensure the quality and safety of agricultural products from the production end. In addition, digital technologies can help improve resource utilization [57], land productivity, and agricultural productivity. Based on agricultural big data technology, both farmers and enterprises can achieve precise management of resources such as land, crops, and agricultural inputs, thereby promoting the efficiency and sustainability of energy production and use. On the management and sales side, the digital transformation of agriculture can realize the precise connection between the production and sales of agricultural products, promote the digitization of all aspects of the production and circulation of agricultural products, improve the efficiency of information acquisition and processing between producers and buyers, and reduce the uncertainty of operations and transactions. At present, some agricultural enterprises have integrated digital technologies, such as big data and cloud computing into the agricultural production process. Through the establishment of a digital pollution monitoring system [58], real-time monitoring and transmission of production conditions can be carried out, and an intelligent expert system can be used to accurately release resources to avoid the resource waste and pollutant discharge, which can significantly improve resource utilization rate and protect land resources and ecological environment. Therefore, the following hypotheses are proposed in this study:

**Hypotheses 2** **(H2):**
*VER can improve the TFP of enterprises by driving digital transformation.*


## 3. The Estimation Strategy

To reduce the bias of empirical research, this study uses a two-way fixed-effects model to study the relationship between environmental regulation and agricultural productivity. At the same time, to avoid the problem of selection bias, this study use the semi-parametric estimation method to estimate the TFP of agricultural enterprises. Use the instrumental variable method to solve endogeneity problems and avoid reverse causality. In further research, the mechanism test method is used to explore the mechanism that affects the relationship between environmental regulation and agricultural productivity.

### 3.1. Principal Component Analysis (PCA)

To reflect the characteristics and laws of the research question as comprehensively and accurately as possible, people often consider using multiple indicators in their research. However, this can lead to complications, redundant metrics, and overlapping information. Therefore, estimates should be made using as few metrics as possible that capture most of the original information.

Principal component analysis is one of the most widely used data dimensionality reduction algorithms, originally proposed by Pearson [59]. It transforms the original data (n-dimensional feature) into a set of linearly independent data of each dimension (k-dimensional feature) through linear transformation, where the k-dimensional feature is a new orthogonal feature, also known as the principal component. In layman’s terms, principal component analysis hopes to use fewer variables to explain most of the variables in the original data, and convert many highly correlated variables in our hands into variables that are independent or uncorrelated from each other to solve linearly related problems.

Firstly, the text mining method is used to crawl the keywords about digitization in the annual reports of agricultural enterprises to construct the digital transformation index. The specific method is: first, collect the annual reports of agricultural enterprises from 2009 to 2019, define the keywords related to digitization, and use Python to crawl the part of the annual report related to the operation status; second, conduct word frequency statistics on the selected keywords and organize them. Identify high-frequency words and construct dimension indicators for enterprise digitalization level evaluation (see Table 1); thirdly, according to the constructed index system and the word frequency of each keyword, using principal component analysis, according to the cumulative variance contribution rate of each component $\frac{\sum_{k=1}^{i}\mu_{i}}{\sum_{k=1}^{p}\mu_{k}}\left(i=1,2,\ldots p\right)$ (here, $\mu_{i}$ represents the variance of the i principal component) and eigenvalues, calculate the digital transformation index of each enterprise.

Figure 1 shows the visual analysis of the digital transformation index of China’s agricultural enterprises in 2011 and 2019. The results show that although the development of digital agriculture in China has improved somewhat in recent years, the overall level of digital agriculture is still not high, and only a few developed large and medium-sized cities develop rapidly. Therefore, the task of accelerating agricultural digital transformation and seeking to improve agricultural productivity is imminent.

### 3.2. Semiparametric Estimation (LP and OP)

The current academic calculation methods for micro-enterprise TFP mainly include the fixed-effect method, GMM method, SFA method, and semi-parametric estimation method. To avoid simultaneity bias and sample selectivity bias, this study uses the LP method of semi-parametric estimation method to measure the TFP of enterprises and uses the OP method to re-measure the robustness test.

First build the following production function:(1)$$lnY_{it}=\theta_{0}+\theta_{1}lnK_{it}+\theta_{2}lnL_{it}+\theta_{3}lnM_{it}+\varepsilon_{it}$$

Among them, Y represents the main business income of company i in year t, K represents the net value of fixed assets of company i in year t, represents a capital investment, L represents the number of employees of company i in year t, represents labor input, and M represents company i, the cash actually paid for purchasing goods and accepting labor services in year t represents the intermediate input, and $\varepsilon_{it}$ represents the random interference item. According to model (6), the TFP of each enterprise can be calculated by using the LP method.

Whether it is the problem of simultaneous selection bias or sample selection bias, the OP method gives a better solution. Therefore, this study uses the OP method to re-measure the TFP of enterprises in the robustness test, and constructs the following production model:(2)$$lnY_{it}=\theta_{0}+\theta_{1}lnK_{it}+\theta_{2}lnL_{it}+\theta_{3}EXIT_{it}+\theta_{4}I_{it}+\varepsilon_{it}$$

Among them, EXIT represents the dummy variable of whether the enterprise will exit the market, I represents the investment in fixed assets, and the rest of the variables are consistent with the above. According to model (7), the TFP of each enterprise can be calculated by using the OP method.

Figure 2 shows the visual analysis of the TFP of Chinese agricultural enterprises in 2011 and 2019. Similarly, the visualizations show similarities between agricultural productivity and trends in digital transformation. On the whole, China’s agricultural productivity has risen, but the average level still lags far behind that of developed countries, and there are also large differences among cities. Accordingly, exploring the way that promotes agricultural productivity is the current primary task.

### 3.3. Two-Way Fixed Effects Model

In a regression model, if some variables in the residual term are related to the explanatory variables, but at the same time have an influence on the explained variables, the estimated coefficients of the explained variables will be biased and inconsistent. The solution is to add these variables to the model, but if there are unobservable cases, then the fixed effects model is used to solve this part of the problem. Generally speaking, the article only adopts the linear regression model of individual fixed effect and time fixed effect, which will lead to the omission of temporal features that do not change with the individual and the individual features that do not change with time, thus making the empirical estimation results more biased. Therefore, based on the comprehensive consideration of time fixed effect and industry fixed effect, this paper uses a two-way fixed effect model to carry out a regression analysis on environmental regulation and enterprise TFP.

To explore the impact of environmental regulation on agricultural productivity, this study constructs the following basic econometric models:(3)$$tfp_{it}=\alpha_{0}+\alpha_{1}ver_{it}+\alpha_{2}E_{it}+year_{t}+ind_{t}+\varepsilon_{it}$$

Among them, $tfp_{it}$ represents the TFP of enterprise i in year t, which is the explained variable of this article, $ver_{it}$ represents the degree of VER of enterprise i in year t, which is the core explanatory variable of this article, $E_{it}$ represents for each control variable, $year_{t}$ represents the annual fixed effect, $ind_{t}$ represents the industry fixed effect, and $\varepsilon_{it}$ represents the random disturbance term.

### 3.4. Instrumental Variable Regression Method (IV)

The relationship between economic variables is complex, interdependent, and even mutually causal. For example, theoretically speaking, TFP, as the main factor reflecting technological progress, reflects the technological level and innovation ability of enterprises. The higher the TFP, the stronger the technological innovation ability of the enterprise, the stronger the ability to create economic value, the greater the willingness and possibility of green production, and the greater the corresponding environmental governance expenditure. This leads to the possibility of inter-causal endogeneity problems in this study.

The instrumental variable method can solve the endogeneity problem in the research well. Generally speaking, the null hypothesis condition is that the explanatory variable is not correlated with the random perturbation term. If there is a violation of this assumption, a variable that is highly correlated with the explanatory variable and not with the random perturbation term is used. Therefore, this study selects the natural logarithm (bjdis) of the geographical distance from each city to Beijing as an instrumental variable, and uses 2SLS to further conduct an endogeneity test to exclude the influence of reverse causality.
(4)$$tfp_{it}=\alpha_{0}+\alpha_{1}ver_{it}+\alpha_{2}E_{it}+year_{t}+ind_{t}+\varepsilon_{it}$$
(5)$$ver_{it}=\alpha_{0}+\alpha_{1}distance_{it}+\alpha_{2}E_{it}+year_{t}+ind_{t}+\varepsilon_{it}$$

Equations (4) and (5) are regression models of 2SLS, Equation (4) is the estimation equation of the second stage, and Equation (5) is the estimation equation of the first stage. The meanings of variables in Equation (4) are the same as those in Equation (3). In Equation (5), distance represents the geographical distance between the city where enterprise i is located and Beijing in year t, and other variables are the same as in Equation (4).

The reason why the independent variable is selected with a lag of one period is that the positive effect of environmental regulation will affect the agricultural productivity of the current period, and there is no “advance effect”. Therefore, there is no necessary connection between the independent variable lagging one period and the dependent variable, which satisfies the conditions of the instrumental variable.

The reasons for selecting this instrumental variable are: first, Beijing is the capital of China and the central city of economic development. The vigorous development of heavy industry in the early days caused serious environmental pollution [60] and waste of resources. Therefore, the Beijing Municipal Government and the People’s Congress have issued a series of laws and regulations on pollution prevention and control to protect the environment, and the degree of environmental regulation is stricter than in other cities. In addition, studies have shown that the richer people’s environmental knowledge, the more active their pro-environmental behaviors are [61]. Beijing has high-quality educational resources and has gathered a large number of high-quality enterprises. The managers of these enterprises and residents have rich environmental protection knowledge and a strong awareness of environmental protection. According to the geographical spillover effect [62] and knowledge spillover effect [63], Beijing has an obvious radiation effect on the environmental regulation behavior of enterprises in neighboring areas, and the radiation effect increases with the geographical distance. The influence of enterprise development is gradually reduced, satisfying the requirement of exclusivity of instrumental variables. Therefore, the closer an enterprise is to Beijing, the more its selection behavior is affected by radiation, and the more likely it is to implement voluntary environmental regulation; Second, since the development of Beijing, high-tech industries and service industries are the main industries, and agriculture accounts for a small proportion of the three major industries. The geographical distance between the cities where enterprises are located and Beijing does not have a great impact on their agricultural productivity. At the same time, the geographic distance from each city to Beijing is objective, and there is no correlation with the explained variables, which can effectively ensure the validity and exogenous conditions of the instrumental variables.

### 3.5. Mechanism Analysis

The traditional mediation effect model cannot uniformly overcome the endogeneity problem of explanatory variables and mediator variables. When implemented, the omitted variables in the endogeneity discussion will be identified as mechanisms, resulting in estimation errors and errors in the conclusion of mediation effects. Therefore, scholars now mostly use a two-step approach for mechanism testing. First, the mechanism and explanatory variables are tested by regression. At this point, the explanatory variables in the study have already found the instrumental variables, and there is no endogenous variable other than the explanatory variables in the regression using the mechanism. Then, theoretically demonstrate the relationship between the mechanism and the explained variables.

Therefore, to demonstrate the impact of digital transformation on the relationship between environmental regulation and agricultural productivity, this study constructs the following mechanism test model:(6)$$digital_{it}=\beta_{0}+\beta_{1}lner_{it}+\beta_{2}E_{it}+year_{t}+ind_{i}+\varepsilon_{it}$$

Among them, $digital_{it}$ represents the digital transformation index of company i in year t, and the rest of the variables are the same as explained above.

## 4. Data and Descriptive Evidence

### 4.1. Data and Sample

The dataset used in this study is based on the balanced panel data from 2011 to 2019, and the selected sample data comes from the Wind and the CSMAR. In addition, the data about digital transformation in this study are all derived from the company’s annual report data collected and obtained manually, and are further determined by text mining and principal component analysis. To ensure the rigor and representativeness of the research, this study selected enterprise data from 2011 to 2019. The reason is that since the outbreak of the new crown epidemic at the end of 2019, the operating conditions of most Chinese companies have been traumatized to varying degrees, especially in agriculture. The specific performance is due to the impact of the epidemic, the inventory of agricultural material dealers and retailers is generally low, and most businesses are suspended, which makes the channels for obtaining agricultural production materials few and far between. In addition, due to the “one-size-fits-all” policy implemented in many areas during the epidemic, there is still great instability in whether farmers can complete spring ploughing, inspection and control of crops, etc., which leads to the decline in output of agricultural products and the problem of insufficient supply. The above phenomena have had a serious impact on farmers and agricultural enterprises, and are the root cause of the vicious circle of the industrial chain of agricultural production. Therefore, this study excluded the data during the new crown epidemic. This study mainly discusses the impact of environmental regulation on agricultural production under the micro-environment. Therefore, this study target the research object in Chinese agricultural enterprises, and control the factors that may affect the explained variables. Correspondingly, it is a relatively stable and comprehensive study. The definitions, calculation methods, and descriptions of key variables are shown in Table 2.

### 4.2. The Descriptive Statistics Analysis

Table 2 reports the definitions, calculation methods, and descriptive statistics of key variables in the empirical study. The explained variable of this study is TFP, which is a key factor used to measure how much technological progress contributes to economic growth. It can be calculated using the semi-parametric estimation method. The core explanatory variable is the intensity of environmental regulation, which is defined as the ratio of corporate environmental governance costs to corporate added value.

In terms of control variables, this study has been controlled by both the enterprise and the government. At the enterprise level, factors such as the scale, age, asset-liability ratio, market value, and equity concentration of the enterprise are all important factors that affect the business activities of the enterprise, and are often used as control variables. At the government level, government subsidies to enterprises may be used to invest in technology, thereby improving TFP, so government subsidies should be controlled in the model.

From the descriptive results in Table 2, it can be seen that the maximum value of the TFP of agribusiness is 10.892, the minimum value is 6.205, and the standard deviation is 0.947, indicating that the Chinese agribusiness productivity is unbalanced, but the gap is not large, and it also shows that the overall development level of enterprises in China is not balanced. The degree of VER of enterprises shows the characteristics of “small mean and large standard deviation”, indicating that the environmental protection awareness of Chinese agribusiness has not been fully popularized, and there are obvious differences in the intensity of environmental regulation among enterprises. Some enterprises ignore the policy requirements of green production and show a negative attitude towards strengthening environmental regulation. From the perspective of the remaining control variables, except for the large differences in enterprise scale, Tobin’s Q value, and government subsidies, the differences in the remaining control variables are relatively small, indicating that the sample selected in this study has strong coverage, and may affect the Numerous factors of the research question are controlled to improve the science and credibility of the research.

## 5. Empirical Results Analysis

### 5.1. Benchmark Regression Results and Discussion

Based on the analysis of the research methods above, to eliminate the influence of heteroscedasticity, this study takes the logarithm of all variables, and then performs regression testing. Table 3 shows the results of the benchmark regression.

Model (1) showed that there was a significant positive correlation between environmental regulation and agricultural total factor productivity. The TFP was 1.8% when the intensity of environmental regulation increased by 1 percentage point. Model (2) controlled for time fixed effects, and the results still showed that environmental regulation was significantly positively correlated with TFP, and the estimated coefficient increased significantly. Model (3) controlled for industry fixed effects, and model (4) controlled for two-way fixed effects. The results showed that environmental regulation was always positively correlated with TFP. This indicated that voluntary environmental regulation had a significant positive promoting effect on the total productivity of agricultural enterprises, which verified hypothesis 1 of this paper. In addition, the stepwise test results also showed that the factors that may affect environmental regulation and its effects were well controlled in this paper, and the control variables selected were scientific and feasible.

### 5.2. Robust Analysis

To verify the robustness of the research conclusions, a robustness test was carried out in this study, and Table 4 reported the results of the robustness test.

First, we replaced the explanatory variable and the explained variable. In model (1), the ratio of the average value of environmental governance expenditure to the value-added of enterprises was used as an alternative explanatory variable. The test results showed that the regression coefficient of environmental regulation was 0.023, which was significant at the 1% level; In model (2), the firm’s TFP was re-measured with the op method. The results showed that the regression coefficient between the re-measured TFP and the original explanatory variable was 0.053, which was significant at the 1% level; In model (3), the surrogate explanatory variables and the explained variables were used for regression, and the regression coefficient was 0.063, which was also significant at the 1% level. The above results were consistent with the benchmark regression results. Second, the interaction fixed effects were added. Controlling the time and industry fixed effects can only consider the homogeneous economic shocks in the time and industry dimensions, resulting in some unobservable heterogeneous shocks being ignored. Therefore, interaction fixed effects needed to be introduced into the model. The regression results of model (4) showed that environmental regulation had a significant positive impact on the TFP of agribusiness, and the regression coefficient was 0.016, which had passed the 5% significance level test, indicating that the regression results were robust; Third, the continuous variables were abbreviated at the 1% and 99% quantiles. To prevent the influence of outliers on the research results, in model (5), the variables were trimmed to smooth the data. The model regression results showed that the regression coefficient of environmental regulation was 0.019, which was significant at the 5% level. To sum up, the results of the robustness test showed that environmental regulation could indeed promote the improvement of the TFP of agribusiness, and the conclusions of this study were quite robust.

### 5.3. Instrumental Variables Estimation Results and Discussions

This section mainly used the instrumental variable method to eliminate endogeneity problems to ensure the stability of research conclusions. In this study, the first-order lag term of the explanatory variable was used as an instrumental variable for regression; secondly, the two-stage least squares method was used for further endogeneity tests to exclude endogeneity problems. The specific test results were shown in Table 5.

First, considering the lag of the impact of environmental regulation, model (1) selected the explanatory variable with a lag of one period as an instrumental variable. The 2SLS regression results showed that there was a significant positive correlation between the explanatory variable lag period and the explained variable, and it was significant at the 1% level. The regression results of the second stage showed that VER had a positive effect on the improvement of the TFP of enterprises, and it was significant at the level of 5%. After controlling for the endogeneity problem, the influence coefficient of environmental regulation had been further improved and maintained a significant level of 5%. In addition, the F value in the first stage was significantly greater than 10, indicating that the selected first-order lag term as an instrumental variable was completely effective.

In addition, the model (2) in Table 5 was regressed based on the selected geographic distance as an instrumental variable. The test results showed that after considering the endogeneity problem, the conclusion that VER had a significant promoting effect on the TFP of enterprises still holds. From the first-stage regression results, it can be seen that there was a positive correlation between environmental regulation and the geographic distance between cities and Beijing, indicating that instrumental variables have better explanatory power for endogenous variables. In addition, the F value in the first stage was significantly greater than 10, which ensured the validity of instrumental variable selection and avoided the problem of weak instrumental variables. The second-stage regression results were consistent with the above research conclusions, rejecting the hypothesis of mutual causal relationship between variables, proving that the research conclusions were robust, and also indicating that the selection of geographic distance as an instrumental variable for environmental regulation was scientific and reasonable.

### 5.4. Analysis of the Mechanism of Action

Based on the “Porter Hypothesis”, environmental regulation will prompt enterprises to carry out technological innovation to make up for the cost of environmental protection, thereby promoting the improvement of TFP. The theoretical analysis above proves that, in the era of Industrialization 4.0, with the continuous development of digital information technology, environmental regulation had accelerated the pace of digital transformation of enterprises, and had also accelerated the arrival of the era of digital agriculture. Therefore, this section will further analyze and discuss how digital transformation affected the relationship between environmental regulation and agricultural productivity. The test results are shown in Table 6.

The test results showed that the regression coefficient of environmental regulation on the digital transformation of agricultural enterprises was 0.02, and it had passed the 1% significance level test. This indicated that environmental regulation accelerated the digital transformation of agricultural enterprises, that is, voluntary environmental regulation behavior of enterprises will induce them to carry out technological innovation, adopt digital production mode, integrate information technology and agricultural production, and promote the development of digital agriculture and smart agriculture. The digital economy represented by digital technology was fully integrated with the agricultural economy, and the latter was more inclusive, improved agricultural productivity by supplementing other production factors to improve efficiency [64], and then promoted the high-quality development of Chinese agriculture and guaranteed national food security. Hypothesis 2 of this paper was verified.

## 6. Conclusions and Further Discussion

### 6.1. Conclusions

Based on the data of Chinese agricultural enterprises from 2011 to 2019, this study calculated the TFP and digital transformation index of each enterprise, and made visual analysis. The results showed that from 2011 to 2019, the digital transformation level and productivity level had improved to a certain extent, but the overall value was not high and the distribution was uneven, especially the status quo of digital transformation [65]. In addition to a few megacities (such as Beijing and Shanghai), the level was relatively high, and the rest of the cities were at a low or medium-low level. However, the enterprises distributed in the Yangtze River basin had a better development situation. This showed that the development of agricultural enterprises in different regions of China was not balanced, and the spatial distribution had a relatively stable trend.

Second, by constructing a two-way fixed effect model, we empirically analyzed the impact of VER on agricultural productivity from the perspective of digital transformation. The results showed that there was a significant positive relationship between VER and TFP. Both the robustness test and the endogeneity test support this conclusion. The results were consistent with the conclusions of previous studies [66]. That is, enterprises voluntarily adopted environmental protection behavior, while it reduced environmental impact and improving productivity. This meant that as agricultural producers and intermediaries provided resources for farmers, the environmental protection behavior of agricultural enterprises had a direct impact on the macro environment and their own development. Voluntary environmental regulation of enterprises was beneficial to improve their awareness of green production [67,68], and then promoted technological innovation and improved production efficiency.

Third, this study was the first to link VER, digital transformation, and agricultural productivity. Mechanism test results showed that the relationship between VER and agricultural productivity was positively transmitted by digital transformation. This indicated that the impact of VER on agricultural productivity would deepen with the speed of enterprise digital transformation. This was the original suggestion of this study. This research result verified and enriched the innovation compensation effect of “Porter hypothesis” to some extent [28]. Therefore, the more active enterprises were in implementing environmental regulation, the more fully motivated they would be to pursue digital transformation and the better the effect of promoting agricultural productivity. However, previous studies have not paid attention to this point. However, it is worth noting that the relationship between VER and agricultural productivity is complex. Digital transformation is not the only way that VER affected agricultural productivity, nor does it necessarily play a positive transmission role in the relationship between any form of environmental regulation and the productivity of any type of industry. This study only confirmed the relationship between the three in the context of digital transformation.

### 6.2. Policy Recommendations

This study analyzed the relationship between voluntary environmental regulation and agricultural productivity from the perspective of digital transformation. The results showed that there was a significant positive relationship between voluntary environmental regulation and agricultural productivity. Mechanism test results developed that voluntary environmental regulation could improve agricultural productivity through digital transformation. Based on the above research conclusions, we put forward the following policy recommendations:

First, reform and improve the rural land system and property rights system to make full and efficient use of cultivated land resources. Through the large-scale farming method, solve the problem of land fragmentation. Based on the characteristics of flat and concentrated contiguous cultivated land, the agricultural infrastructure should be improved and the level of farming mechanization should be improved, so as to improve the land utilization rate and reduce agricultural environmental risks.

Second, China is one of the countries that uses the most chemical fertilizers and pesticides, and it is also one of the countries with the most serious soil pollution. At the present stage, the application form of soil fertilizer should be optimized, and the application rate of organic fertilizer should be increased appropriately, so as to form a fertilization structure with organic fertilizer as the main fertilizer and chemical fertilizer as the supplement. In addition, we should give full play to the advantages of digital agriculture, carry out land science and technology innovation, and establish and improve a land science and technology innovation system in combination with farmland protection, green development and ecological restoration, so as to ensure sustainable land use. Such as artificial intelligence, big data accurate algorithm characteristics of digital technology, realize accurate monitoring of crop and soil, and the different crops in different growth stages of pollution stage differentiation of fertilizing land planning, achieve the goal of precise fertilization, scientific management, improve land quality, from the overall improve agricultural ecological environment and realize the sustainable utilization of land.

Third, in the process of preventing and controlling agricultural pollution and realizing sustainable land development, the government should play the role of the main force. On the one hand, we should rely on relevant government departments to strengthen publicity and guidance, and strengthen people’s awareness of green production and pollution prevention and control through special lectures and seminars, so as to promote the formation of sustainable soil management mechanism, and implement the concept of sustainable land development. On the other hand, government departments should restrict the unreasonable behavior of agricultural producers by compulsory means, improve the relevant laws and regulations of land, and restrict the behavior of soil destruction such as excessive development of land and excessive use of chemical fertilizers and pesticides by legal means, so that the development and utilization of land are completely under standardized management. In addition, the concept of sustainable utilization is put into regular management, so as to comprehensively improve the quality and efficiency of pollution control.

### 6.3. Limitations and Further Research

This study also has certain research limitations. Firstly, since the environmental governance of Chinese enterprises has not been fully disclosed to the public, the amount of environmental regulation data obtained is relatively small. It is necessary to obtain more accurate data in the case of further improving the information disclosure system in the future to improve the index system, to ensure the generalizability of research findings; Secondly, this study discusses the impact of VER on Chinese agribusiness. Based on this study, the follow-up research can introduce different types of environmental regulation behaviors in own countries and different countries with better environmental performance for comparative research, to provide more accurate and practical policy recommendations for China’s agricultural development.

## Figures and Tables

**Figure 1 ijerph-19-10794-f001:**
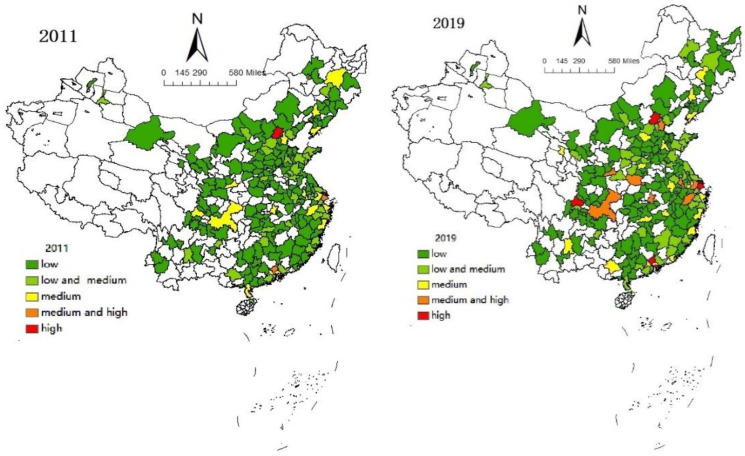
Trends in agricultural digital Transformation from 2011 to 2019.

**Figure 2 ijerph-19-10794-f002:**
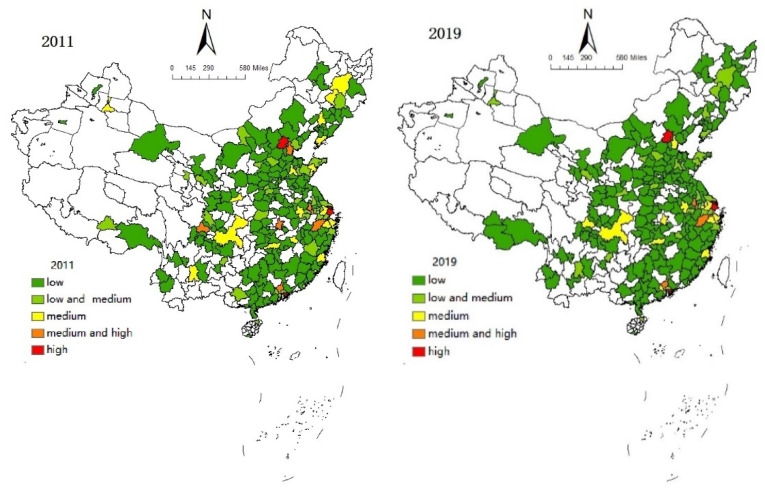
Trends in agricultural productivity from 2011 to 2019.

**Table 1 ijerph-19-10794-t001:** Index system of digitalization level.

	Dimension	High Frequency Keywords
Digital transformation	Application of digital technology	Digitization, digital marketing, digital technology, digital technology, digital operation, digital terminal, digital economy, digital trade, digital system, digital supply chain, information age, informatization, informatization technology, information integration, information communication, information integration, computer technology
	Smart business	Internet of Things, edge computing, cloud computing, cloud services, cloud, automation, 5G, smart age, smart construction, smart business, intelligence, 3D printing, 3D technology, 3D tools, AI, robotics, machine learning
	Internet business model	Network, Internet, digital currency, blockchain, e-commerce, cross-border e-commerce, e-commerce platform, electronic technology, electronic technology, online, online and offline, B2B, O2O, C2C, P2P, C2B, B2C
	digital information	Data integration, data fusion, data information, data management, data assets, big data

**Table 2 ijerph-19-10794-t002:** Descriptive statistics.

Variables	Definition and Measurement	N	Mean	Str	Min	Max
lNTFP	Semiparametric estimation method (described above)	1197	8.202	0.947	6.205	10.892
VER	Ln(Environmental treatment costs/corporate value added)	1197	3.435	1.550	−1.828	7.801
Size	Ln(Total assets at the end of the year)	1197	22.166	1.159	19.977	25.765
Lev	Total liabilities/Total assets	1197	0.433	0.194	0.051	0.960
Age	Ln (current year − the year of listing)	1197	2.122	0.704	0.693	3.219
ROA	Average value of net profit at the beginning and end of the period/Average value at the beginning and end of the total assets period	1197	0.034	0.053	−0.232	0.322
TobinQ	Company market value/Net assets	1197	1.892	1.094	0.809	9.386
Con	Ln (Shareholding ratio of the largest shareholder + 1)	1197	3.498	0.404	2.178	4.425
Government	Ln (Government grant funds)	1197	15.923	1.769	8.517	20.130
Board	Ln (Board size + 1)	1197	2.240	0.156	1.792	2.565

**Table 3 ijerph-19-10794-t003:** Benchmark regression results.

	Benchmark Regression			
	(1)	(2)	(3)	(4)
VER	0.018 ** (2.30)	0.022 *** (2.68)	0.017 ** (2.02)	0.020 ** (2.50)
Control Variable	YES	YES	YES	YES
Year	NO	YES	NO	YES
IND	NO	NO	YES	YES
N	1197	1197	1197	1197
*R* ^2^	0.795	0.804	0.796	0.807

Note: *** *p* < 0.01, ** *p* < 0.05 indicate that the test result is significant at the 1%, 5% levels, respectively. In parentheses is the absolute value of the *t*-statistic.

**Table 4 ijerph-19-10794-t004:** Robustness test results.

	Robustness Check				
	(1)	(2)	(3)	(4)	(5)
VER		0.055 *** (5.74)		0.019 ** (2.33)	0.021 *** (2.56)
VER2	0.027 *** (3.03)		0.063 *** (6.12)		
Control Variable	YES	YES	YES	YES	YES
Year	YES	YES	YES	YES	YES
IND	YES	YES	YES	YES	YES
Year × IND	NO	NO	NO	YES	NO
N	1167	1197	1167	1197	1197
*R* ^2^	0.815	0.456	0.466	0.808	0.812

Note: *** *p* < 0.01, ** *p* < 0.05 indicate that the test result is significant at the 1%, 5% levels, respectively. In parentheses is the absolute value of the *t*-statistic.

**Table 5 ijerph-19-10794-t005:** Endogenous test results.

	(1)		(2)	
	First Stage	Second Stage	First Stage	Second Stage
L. VER	0.744 *** (25.49)			
VER		0.026 ** (2.10)		0.186 ** (2.09)
BJDIS			0.150 *** (3.84)	
Control Variable	YES	YES	YES	YES
Year	YES	YES	YES	YES
IND	YES	YES	YES	YES
N	898	898	1002	1002
*R* ^2^		0.813		0.766

Note: *** *p* < 0.01, ** *p* < 0.05 indicate that the test result is significant at the 1%, 5% levels, respectively. In parentheses is the absolute value of the *t*-statistic.

**Table 6 ijerph-19-10794-t006:** Mechanism test results.

	Mechanism Inspection
	Digital Transformation
Digital	0.020 *** (6.42)
Control Variable	YES
Year	YES
IND	YES
N	1188
*R* ^2^	0.206

Note: *** *p* <0.01 indicate that the test result is significant at the 1% level, respectively. In parentheses is the absolute value of the *t*-statistic.

## Data Availability

Not applicable.

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
