# Peer review of "The Impact of Environmental Regulation on Agricultural Productivity: From the Perspective of Digital Transformation"

_ijerph, 2022, doi:10.3390/ijerph191710794_

Round 1

Reviewer 1 Report

Comment 1: The authors should discuss motivation for choice of variables and the big question on the need for digitalization in Chinese agricultural sector should be properly highlighted in the introduction and few sentences in the abstract

Comment 2: The objective of the paper presented needs more clarification to suit reader to understand the digital transformation perspective on the impact of environmental regulation on agricultural productivity. Specifically, three paragraphs were written to discuss agricultural production impact on the environment and only just began to write on digital transformation in line 80, without necessarily linking the previous paragraphs to the next sentences. Also, i will suggest that the discussion on the digital transformation perspective should begin in a new paragraph.

Comment 3: The abbreviation VER and TFP was first mentioned in page 3 line 127 without reference to their full form (except in abstract).

Comment 4: The literature is very well written, with the relationships between the variables well established and relevant studies included.

Comment 5: The methodology of the study is well represented and data analysis well documented.

Comment 6: The discussion of the result is okay but can be improved. The authors need to discuss the finding in relation to previous studies that support or oppose their finds.

Comment 7: The conclusion should be subtitled to conclusion, policy recommendation (theoretical and practical implication of the study), and limitation/future research recommendation Conclusion should portray philosophical bearings from what is obtained or discovered in the empirical findings of the study i.e., it should not be a summary or an exposure of the limitations of the study and further areas of research. But the conclusion of the study should be something with an overall thought from the authors objective analysis on why and how things exist or went the way they were discovered.

Reviewer 2 Report

In this paper the Authors describe a model and an instrumental variable method to examine the impact of environmental regulation on agricultural total factor productivity.

The manuscript is interesting and fits well with the aim of the International Journal of Environmental Research and Public Health.

The reviewer recommends improvement in the following items:

(1)   In paragraph 2, the Authors refer to VER and TFP. They should explain what are these parameters and/or if they are acronyms. In this case, the first time they are mentioned, they should be written in their entirety form.

(2)   At the end of Sections 2.1 and 2.2 the Authors make assumptions H1 and H2, but written in this form they are not very clear. The reviewer suggests writing both assumptions in the introduction section and reiterate or make them explicit in Sections 2.1 and 2.2.

(3)   In the formula on line 260, ? appears. The Authors should indicate what this parameter denotes.

(4)   Figures 1 and 2 are unclear. The reviewer suggests using a color map to improve readability.

(5)   The footnotes in Tables 3, 4, 5, and 6 are not clear. Please clarify what they refer to.

According to what said above, the reviewer’s opinion is that the manuscript can be accepted for publication after the described minor revisions.
